# A Vicious Cycle of Osteosarcopenia in Inflammatory Bowel Diseases—Aetiology, Clinical Implications and Therapeutic Perspectives

**DOI:** 10.3390/nu13020293

**Published:** 2021-01-20

**Authors:** Dorota Skrzypczak, Alicja Ewa Ratajczak, Aleksandra Szymczak-Tomczak, Agnieszka Dobrowolska, Piotr Eder, Iwona Krela-Kaźmierczak

**Affiliations:** Department of Gastroenterology, Dietetics and Internal Diseases, Poznan University of Medical Sciences, 49 Przybyszewskiego Street, 60-355 Poznan, Poland; alicjaewaratajczak@gmail.com (A.E.R.); aszymczaktomczak@ump.edu.pl (A.S.-T.); agdob@ump.edu.pl (A.D.);

**Keywords:** sarcopenia, IBD, Crohn’s disease, ulcerative colitis, malnutrition, low muscle mass, low muscle strength, nutritional therapy

## Abstract

Sarcopenia is a disorder characterized by a loss of muscle mass which leads to the reduction of muscle strength and a decrease in the quality and quantity of muscle. It was previously thought that sarcopenia was specific to ageing. However, sarcopenia may affect patients suffering from chronic diseases throughout their entire lives. A decreased mass of muscle and bone is common among patients with inflammatory bowel disease (IBD). Since sarcopenia and osteoporosis are closely linked, they should be diagnosed as mutual consequences of IBD. Additionally, multidirectional treatment of sarcopenia and osteoporosis including nutrition, physical activity, and pharmacotherapy should include both disorders, referred to as osteosarcopenia.

## 1. Introduction

Sarcopenia is a syndrome associated with a reduction of skeletal muscle mass, leading to decreased muscle strength [1]. In the past, sarcopenia was thought to be a part of ageing, specifically for older people. However, sarcopenia may also affect the young population. There are many sarcopenia risk factors, such as ageing, in fact, sarcopenia is characteristic for older people, although it may occur at all stages of life, including childhood and adolescence [2,3,4]. Furthermore, other risk factors for sarcopenia comprisethe physiological and genetic factors, e.g., disorders involving metabolic pathway of activin A-myostatin-follistatin or mutation [5] of *PRDM16* gene [6], nutrition factors, such as dairy consumption, the intake of calcium, vitamin D, phosphate [7], as well as malnutrition [8]. Important elements of sarcopenia development are coexisting diseases, such as inflammatory bowel diseases (IBD) [8], chronic kidney disease [9], obesity [6], frailty syndrome [10] and others. In addition, physical activity and ethnicity are also significant [1,10].

The first definition of sarcopenia was based on a low muscle mass. In 2010, the European Working Group on Sarcopenia in Older People (EWGSOP) extended this definition and added information regarding the muscle function (strength and performance). [11]. In 2018, the description was updated again by EWGSOP2 and was defined as a progressive, systemic skeletal muscle disease, leading to an increased probability of adverse events, such as falls, fractures, physical disabilities, and death [2].

Sarcopenia includes various types which are classified according to the etiology. The primary sarcopenia is associated with ageing without other factors. Nevertheless, certain factors may cause secondary sarcopenia where it is possible to identify sarcopenia dependent on physical activity (e.g., sedentary lifestyle, zero gravity), sarcopenia associated with other conditions (e.g., damage to the internal organs, inflammatory diseases, cancer, endocrine diseases), sarcopenia associated with nutrition (insufficient supply of energy, protein, malabsorption, gastroenterology disorders, medications). In general, sarcopenia is diagnosed on the basis of a low muscle mass (criterion 1) and muscle quality or quantity (criterion 2). Additionally, severe sarcopenia is diagnosed if the performance of the muscles is low (criterion 3) [11].

The diagnostic algorithm of sarcopenia is based on the F-A-C-S pathway (find cases-assess-confirm-severity) (Table 1) [12].

Skeletal muscle mass is assessed by means of various techniques, such as the DXA evaluation of appendicular skeletal muscle mass (ASMM), MRI (magnetic resonance imaging) assessment of the total skeletal muscle mass (SMM) and ASMM, as well as MRI or CT (computed tomography) evaluation of the midthigh muscle or lumbar muscle cross-sectional area. The quality of muscle may be assessed by means of CT in the midthigh, as well as by using MRI or CT in the total body [2].

Osteoporosis is defined as a systemic metabolic skeletal disease characterized by progressive bone mass loss, decreased bone structure and susceptibility to fracture [17,18]. Depending on the location, osteoporosis can be classified as either local, concerning a particular skeletal fragment, or generalized, covering the entire skeletal system. Generalizedosteoporosis can be either primary (80% of cases) or secondary (20% of cases). Primary osteoporosis involves an involuntary and idiopathic form, affecting children, adolescents and middle-aged adults. In involuntary osteoporosis, type I and II are distinguished. Type I comprises postmenopausal osteoporosis (80% of cases) and type II is senile osteoporosis, which affects people of both sexes after 70 years of age (20% of cases).Secondary osteoporosis is a result of bone mass loss due to the presence of certain diseases or the use of medications affecting bone metabolism. According to the World Health Organization (WHO) and the International Osteoporosis Foundation (IOF), the basis for the diagnosis is the measurement of bone mineral density (BMD) in the lumbar spine and in the proximal end of the femur by means of dual-energy X-ray absorptiometry (DEXA), based on T-score, expressed as the number of standard deviations, with peak bone mass as the reference point.The following diagnostic criteria were adopted: T-score > −1 SD—normal value, T-score from −1 to −2.5 SD—osteopenia, T-score < −2.5 SD—osteoporosis, T-score < −2.5 SD and osteoporosis fracture—advanced osteoporosis. In young adults and children it is recommended to include Z-score, where the reference point is a group of peers [18,19,20].

According to the estimates, over 200 million people in the world suffer from osteoporosis [21]. In 2010, by means of employing the WHO diagnostic criterion, it was determined that 22 million women and 5.5 million men in the EU had osteoporosis [22].

Osteoporosis is a disease of a complex etiology—a strong multigenetic background is additionally modified by numerous environmental factors.Genes controlling ossification and proper maintenance of bone structure belong to four groups of biological factors: cytokines (Il-1, Il-6), transforming growth factors (TGFbeta1(transforming growth factor), TNF), matrix components (COLIA1, COLIA2, BGP) and calcitropic hormone receptors (VDR, ESR) [23,24].

The main environmental risk factors include age, female gender, ethnic group (osteoporosis is more prevalent in the white and yellow ethnicities), coexisting diseases (endocrinopathies, diabetes, gastroenterological, rheumatic, and kidney diseases), low-calcium and low-protein diet, vitamin D deficiency, excessive coffee consumption, malnutrition, lack of exercise, smoking, alcohol abuse, medications (glucocorticosteroids, antiepileptic medications) [25,26].Osteoporosis is called “silent bone eater” due to its prolonged asymptomatic course. The clinical manifestation of osteoporosis is bone fractures—mainly of the femoral neck, vertebrae, ribs, humerus, radius or tibia. The fractures usually occur as a result of minor injuries, and thus are referred to as low energy fractures. The first and the most frequent vertebral fractures occur in the course of osteoporosis, most often in the thoracic and lumbar sections. However, the most dangerous consequences of osteoporosis are femoral neck fractures, resulting in more than 50% of patients losing their ability to move freely [18,27]. It is widely recognized that osteoporosis and the resulting fractures are also associated with an increased mortality [27]. In the case of femoral neck fracture, most deaths occur during the first 3−6 months following the fracture [28].

## 2. Bone and Muscle Tissue

In recent years, muscle and bone tissues arehave become inextricably intertwined [29]. In fact, communication of these tissues by paracrine and hormonal signals may lead to the stimulating development and to the adaptation related to strains and microtrauma occurring since the embryonic period to old age [30]. Furthermore, numerous risk factors are mutual for sarcopenia and osteoporosis, including mechanical, biochemical, hormonal, genetic, as well as factors associated with a lifestyle which lead to the involution of the bone−muscle unit [29].

Physical activity and mechanical strain play a role in the pathogenesis of both sarcopenia and osteoporosis, which may indicate an interaction between both tissues. A good example of such interaction is bone mechanostat theory, which refers to the fact that muscle affects bone by applying certain force, whereas the threshold decides about the formation or resorption of bone [31]. Moreover, an increase in muscle mass may stimulate bone growth by influencing collagen fibers.

Muscle tissue produces a number of myokines which may affect bone [32]. Myokines participate in bone metabolism and additionally fulfil physiological functions, i.e., they participate in muscle glucose metabolism and influence muscle vascularity [33]. According to Kaji, the most vital myokines include myostatin, transforming growth factor-β, bone morphogenetic proteins (BMP), activin, follistatin, insulin-like growth factor 1 (IGF-1), fibroblast growth factor (FGF-2), osteoglycan, family with sequence similarity 5, member C (FAM5C), irisin, interleukins (IL, among otherIL-6, IL-7, IL-15), monocyte chemoattractant protein (MCP-1), matrix metallopeptidase (MMP-2) as well as osteonectin [32].

The myostatin belongs to the TGF-β superfamily and inhibits skeletal muscle. This protein activates the activin-like kinase (ALK) by binding to the activin receptor type IIB (ACV2B) which leads to the phosphorylation of SMAD protein. Research shows that myostatin affects bone mineral density (BMD) through the inhibition of osteoblastogenesis and by increasing osteoclastogenesis [34]. A GWAS (genome-wide association studies) study indicated that the gene coded myostatin (*GDF8* gene, growth differentiation factor 8) is mutual for the pathogenesis of osteoporosis and sarcopenia. Moreover, Zhang et al., reported that polymorphisms of the myostatin gene affect BMD peak in Chinese women [35].

Important factors which influence bone formation comprise BMP and TGF-β, participating in the growth and the differentiation of osteoblasts and chondroblasts [36]. This stimulation may be inhibited by activin (ligand for the myostatin receptor), which additionally increases osteoclastogenesis [32].

Irisin is produced by the skeletal muscles following activity. It affects bone tissue by a pathway of Wnt-β-katenin, p38 MAPK (mitogen-activated protein kinases), and ERK (extracellular signal-regulated kinases) influencing osteoblasts differentiation. Additionally, irisin inhibits osteoclastogenesis by affecting RANKL (Receptor Activator for Nuclear Factor κ B Ligand) and NFATc1 (nuclear factor-activated T cells c1) pathway [37].

Cytokines produced by muscles also constitute factors that involve bone tissue. One of the main interleukines is IL-6, which is produced in a response to a muscle spasm. In fact, IL-6 has been associated with an increased activity of osteoclasts and bone resorption [38]. Moreover, IL-7 and IL-15 also affect bone tissue.

IGF-I and FGF, the expression of which has been found in muscles, are crucial for bone development and may affect BMD. IGF-I stimulates bone remodeling, influencing osteoblasts, and osteoclasts [39]. On the other hand, FGF from muscle tissue is produced in a response to microtrauma of miotubules, and may stimulate bone formation and regeneration [40].

## 3. Sarcopenia in Patients Suffering from IBD

Sarcopenia is common in patients suffering from IBD [41,42,43]. The pathogenesis of muscle loss is multifactorial, and includes ageing, systemic inflammation, mitochondrial disorders, increase proteolysis, and insulin resistance [41]. Thus, sarcopenia constitutes a predictive factor for surgical intervention in IBD patients. Moreover, it is associated with an increased risk of severe postoperative complications (more blood transfusions, more frequent hospitalizations at the ICU, and more frequent perioperative sepsis and venous thrombosis) [44,45].

Contrary to its appearance, sarcopenia is not simply correlated with BMI (Body Mass Index) in patients suffering from IBD. Therefore, BMI is not an appropriate marker. In the sarcopenic patients suffering from IBD, BMI is often normal or indicates overweight or obesity (sarcopenic obesity) [43,46,47]. Hence, additional simple methods of muscle strength evaluation may be used which include the hand-grip strength test and the walking speed test recommended by EWGSOP2 [2,42,43]. Moreover, hand-grip strength assessment using a dynamometer should be used for the determination of a low muscle mass in IBD patients, which allows for the diagnosis of sarcopenia and constitutes a predictive factor of osteopenia and osteoporosis [42]. The occurrence of sarcopenia is strongly associated with osteopenia and osteoporosis in young patients suffering from IBD. In fact, both sarcopenia and osteoporosis may be dependent on common mechanisms, such as low physical activity, oxidative stress and chronic inflammation [48].

The population of children with IBD present significant muscle and bone tissue insufficiency, as well as growth deficiency [3,49,50,51]. The mechanism of sarcopenia development in pediatric patients is not well known, although it may be associated with a decreased intake of nutrients (including vitamin D) [52], low physical activity, and hypermetabolism [3]. Furthermore, it is possible that a loss of muscle mass leads to bone disorders among children [50]. In fact, poor body weight gain, lower height, a decreased bone development, and late adolescence in children result from direct factors, such as inflammation, as well as from indirect factors, i.e., low energy intake, malnutrition, an increased energy demand, and malabsorption [49,50]. It is difficult to assess which mechanism is more important for the development of sarcopenia—the disease itself or the treatment (glucocorticosteroids-GCs, anti-TNFα, and others). It is vital to notice that GCs and anti-TNF-α may influence muscle tissue [4,49]. However, the animal study indicated that only nutritional intervention and anti-inflammation treatment are necessary for optimal growth [53]. According to Altowati et al., aone-year therapy with anti-TNFα leads to clinical improvement and an increase in osteoblast activity. Nevertheless, it does not improve bone and muscle tissue disorders, or hand-grip strength [4].

The occurrence of sarcopenia is varied and depends on numerous risk factors shown in Table 2 [54].

## 4. Osteoporosis in Patients Suffering from IBD

Patients with IBD belong to the group presenting a higher risk of low bone mineral density (BMD) and osteoporosis which are both multifactorial disorders [60,61].The risk factors of osteoporosis are presented in Table 3. Osteoporosis increases the risk of pathological fractures [62] which may cause a decreased quality of life. It is vital to bear in mind that osteoporosis affects not only the elderly, but also young people suffering from IBD [63]. Patients suffering from IBD should be screened for osteoporosis if they comply with one or more risk factors, such as postmenopausal females, males aged above 50 years, hypogonadism, as well as a history of vertebral fractures and chronic steroid use. Patients with a low trauma fracture or osteoporosis need additional tests (including serum concentration of calcium 25(OH)D, alkaline phosphatase) [64].

Only 43% of IBD patients presented a normal BMD in the lumbar spine. Osteopenia occurred in 46%, whereas osteoporosis affected 11% of patients [84]. On the other hand, Noble et al., demonstrated that osteoporosis occurred in 15% and 16% of patients with ulcerative colitis (UC) and Crohn’s disease (CD), respectively. Additionally, low BMI was associated with a decreased BMD [85] and the BMD of patients with IBD was significantly lower than in healthy individuals [86]. It is vital to notice that osteoporosis and osteopenia occurred more frequently in patients with CD than with UC [48]. According to Schneider et al., the CD patients with sarcopenia presented a lower BMD when compared with the patients without sarcopenia. Additionally, 91% of patients with sarcopenia additionally suffered from osteoporosis. In fact, skeletal muscle mass is correlated with BMD [48]. Therefore, sarcopenia and a low free fat mass may be predictors of osteopenia and osteoporosis in IBD patients [42]. Hence, it might seem that sarcopenia, which is associated with decreased muscle mass and strength, predisposes to falls and, therefore, to bone fracture. However, Harris et al., reported that the relative risk of bone fracture in women with and without sarcopenia remained the same if BMD was within the normal range [87].

## 5. Pharmacological Treatment in IBD and Risk of Sarcopenia and Osteoporosis

### 5.1. Glucocorticosteroids

Glucocorticosteroids are commonly usedin the chronic treatment of patients with IBD. GCs affect bone and muscle tissues [60,65] by means of increasing apoptosis and decreasing osteoblastogenesis and promotion of osteoclastogenesis. Additionally, GCs affect calcium balance by decrease intestine absorption and renal resorption [88,89] and increase the differentiation and activation of osteoblasts which leads to a decrease in BMD [90]. Moreover, glucocorticosteroid therapy decreases muscle mass due to stimulating myostatin expression [9]. As animal studies showed, deletion in the myostatin gene (*GDF8*) protects from atrophy of skeletal muscle generated by GCs. Nevertheless, the effect of GCs is dependent on the dose and treatment duration [91,92].

### 5.2. Thiopurine

Thiopurine is a group ofantimetabolic and immunomodulatory medications, including azathioprine (AZA), mercaptopurine (6-MP), and thioguanine (6-TG). Thiopurine enhances the effect of anti-TNFα and may be used in the treatment of patients suffering from IBD with GC resistance [93,94]. IBD patients are more likely to experience bone mass loss and fractures which stem from various causes (chronic inflammation, malnutrition, GCs therapy). In fact, one animal study indicated that AZA negatively affected the microarchitecture of trabeculae [95]. Furthermore, according to Lee et al., administering thiopurine to IBD patients affects fatigue which stops after the withdrawal of the medication [96]. Moreover, Gupta et al., reported that the treatment with thiopurine had low efficacy in pediatric patients with CD and, simultaneously, negatively affected anthropometric parameters, e.g., free fat mass [97].

### 5.3. Biopharmaceuticals

Biopharmaceuticals, used in the treatment of IBD (infliximab, adalimumab, and certolizumab) are medicines which counteract TNF-αwhich is one of the inflammation markers. The effect of biopharmaceuticals on osteoporosis, as well as on the development and the course of sarcopenia is not well known, and data are mutually exclusive. In contrast, some studies reported that biopharmaceuticals improve BMD and bone tissue parameters [98,99]. However, there is also evidence ofa lack of impact on BMD [100] and other research studies presented data with regard to reducing BMD following treatment [101]. According to Holt et al., patients with myopenia are more likely to experience a loss of response toanti-TNFα than patients with a normal muscle mass [102].

## 6. Sarcopenia and Nutrition in IBD Patients

One of the risk factors of sarcopenia is inappropriate diet, although not much has been established up to date. Nevertheless, it is possible that a “more healthy” diet (including more vegetables and fruits, whole grain products, fish, lean meat, low-fat dairy, nuts, and olive oil) positively affects muscle strength [103].

## 7. Malnutrition

Malnutrition due to insufficientenergy supply is associated with sarcopenia. The deficiency of energy leads to catabolism of fat and muscle tissue in order to obtain the energy necessary for the proper functioning of organs and muscles [104,105]. The study indicated that in patients with overweight or obesity a very low-calorie diet decreased the fat tissue mass regardless of the protein intake percentage [106]. According to Beaudart et al., nutritional status is strongly associated with sarcopenia which occurs significantly more frequently in malnourished patients [107] and it was diagnosed in 77% patients experiencing malnutrition [108].

A particular group with a higher risk of malnutrition comprises patients suffering from IBD. Moreover, although malnutrition affects about 16% of patients, more than 77% of them avoid certain products in order to prevent disease exacerbation [109]. On the other hand, according to Mijac et al., malnutrition may occur in 25.0–69.7% of patients, depending on the method of diagnosing malnutrition [110]. Malnutrition affects IBD patients significantly more frequently when compared to healthy individuals [111]. The pathogenesis of malnutrition among patients suffering from CD or UC is multifactorial and includes such factors as insufficient intake of energy and nutrients, malabsorption, restrictive surgeries, and inflammation [112].The daily intake of calories, protein, fat and vitamin D seems to be particularly relevant for the risk of malnutrition and the development of osteosarcopenia, as shown in Table 4.

### 7.1. Protein

Insufficient protein intake constitutes a risk factor of the development of sarcopenia, since the supplementation of protein improves muscle anabolism which protects from muscle mass loss. A meta-analysis demonstrated that dairy protein intake increased limb muscle mass, although it did not change handgrip and leg press [116]. According to Rondanelli et al., whey protein supplementation, which is an essential amino acid (including 4g leucine), as well as vitamin D in combination with physical activity increased free fat mass, relative skeletal muscle mass, and handgrip strength when compared to the control group [117]. Additionally, the supplementation of beta-hydroxy-beta-methylbutyrate, leucine, and the essential amino acids increased muscle mass and strength, particularly in combination with resistance training [118]. According to Cramer et al., the supplementation of oral nutritional supplements (ONS) including protein (14 g) and vitamin D (147 IU), such as containing protein (20 g), vitamin D (499 IU) and calcium β-hydroxy-β-methylbutyrate (CaHMB) (1.5g) improved muscle strength in malnourished adults with sarcopenia [119]. Moreover, high (≥1.0 g/kg/day) and very high (≥1.2 g/kg/day) intake of protein was associated with higher mobility in the lower extremities when compared with a low protein intake (≥1.2 g/kg/day) [120]. What is more, the supplementation of vitamins D and E increased relative muscle mass, muscle strength, and anabolic markers, such as IGF-1 and IL-2 [121]. Additionally, in older people suffering from sarcopenia, the supplementation of whey protein decreased the skeletal muscle mass, irrespective of whether it was supplied before or after the training [122]. According to Chanet et al., 6-week long consumption of breakfast enriched with the medical product containing whey protein (abundant in leucine) and vitamin D stimulated the synthesis of muscle protein and increased muscle mass in men [123]. The recommendation for older people with sarcopenia includes daily protein intake in quantities of 1.2–1.4 g/kg body weight [113].

Opstelten et al., demonstrated that patients suffering from IBD consumed more animal protein than the healthy respondents [124]. On the other hand, the intake of protein was not significantly different between men with UC in remission and the healthy individuals [125]. Moreover, protein intake among children with IBD was higher when compared to the healthy patients [126]. However, pediatric patients with UC, who did not presentwith CD, consumed a lower amount of protein than their healthy peers [127]. Although studies indicate no significant differences in the protein intake between patients with IBD and the healthy individuals, the supply of these nutrients may not be sufficient for UC and CD patients due to their increased demand. According to the recommendations of ESPEN (European Society for Clinical Nutrition and Metabolism), in the exacerbation of IBD, a supply of protein should be estimated at 1.2–1.5 g/kg body weight [128], thus, this recommendation bears a similarity to the recommendation for patients suffering from sarcopenia.

### 7.2. Vitamin D

The impact of vitamin D on muscle strength and mass remains unclear. However, the occurrence of vitamin D receptor (VDR) in the cells of skeletal muscle was investigated where the presence of VDR would indicate a direct influence of vitamin D on myocytes. Additionally, vitamin D affects the proliferation and differentiation of muscle cells and inhibits expression of myostatin [115].

Individuals with lower serum levels of 25(OH)D presented a lower muscle mass, strength and function than people with a high concentration of 25(OH)D [129]. The meta-analysis indicated that vitamin D supplementation in combination with the resistance training increased the strength of limb muscles when compared to the training without the supplementation [130]. According to Okubo et al., a low level of vitamin D constituted an independent factor associated with sarcopenia. Additionally, a concentration of 25(OH)D3 was positively correlated with the grip strength, as well as the skeletal muscle mass index [131]. Moreover, the frequency of obesity, sarcopenia, and sarcopenic obesity was higher in individuals with insufficient vitamin D intake [132]. Among children under 13 years of age, sarcopenia occurred more frequently in patients with a suboptimal serum level of 25(OH)D (<50 nmol/L) [52]. According to Takeuchi et al., in older adults a supplementation of branched-chain amino acids and vitamin D in combination with the resistance training significantly increased grip strength and calf circumference in comparison with a group with the training, but without the supplementation [133]. However, according to the International Clinical Practice Guidelines for Sarcopenia, evidence to determine the scheme of vitamin D supplementation in older people suffering from sarcopenia is insufficient [134].Following the recommendations, the intake of 50,000IU vitamin D weekly is safe [135]. On the other hand, some studies reported that for adults and the elderly with normal body weight the tolerable upper limit of vitamin D dose amounts to 4000 IU/day, whereas for the obese the dose is equal to 10,000 IU/day [136]. It is vital to notice that vitamin D toxicity is rare and the concentration of 25(OH)D has to be over 150 ng/mL (375 nmol/L) for the symptoms to occur [137].

Patients suffering from IBD are a group with an increased risk of vitamin D deficiency. A serum level of25(OH)Dlower than 50 nmol/L affected 53% and 44% of patients with CD and UC, respectively [138]. Additionally, the concentration of 25(OH)D was significantly lower in IBD patients when compared to the healthy subjects. Both in a severe and moderate period ofUC and CD, the serum level of vitamin D was lower than in remission [139]. According to Głąbska et al., the intake of vitamin D was not different between UC men and the healthy individuals [125]. It is vital to bear in mind that 67% of patients suffering from IBD have low exposure to sunlight, which is essential for the synthesis of vitamin D in the skin [140]. Nevertheless, Lund-Nielsen et al., indicated a lack of impact of vitamin D deficiency on IBD development [141].

### 7.3. N-3 Fatty Acids

The mechanism of the impact of n-3 fatty acids (n-3 FA) on muscle tissue is unclear. It is possible that it affects the anabolic and katabolic pathway, including the synthesis and degradation of protein [114]. Moreover, Yoshino et al., suggested that the supplementation of n-3 FA influenced muscle transcript [142]. However, studies on the supplementation of n-3 FA are ambiguous.

According to Smith et al., a six-month supplementation of n-3 FA significantly increased thigh muscle volume, grip strength, and 1-RM (one repetition maximum) muscle strength in comparison to the placebo group [114]. On the other hand, a 12-week supplementation of n-3 FA did not change muscle mass, grip strength, or the results of TUG (“timed up and go”) among older people [143]. However, the consumption of n-3 FA was associated with muscle mass in hemodialysis patients. Moreover, a high n-6/n-3 ratio was connected with decreased muscle mass [144]. Rolland et al., have shown that a low dose of n-3 FA (800 mg DHA and 225 mg EPA), supplemented alone or combined with a change of lifestyle (including starting resistance training), did not alter muscle strength [145]. It is vital to notice that supplementation of n-3 FA in combination with resistance training did not change muscle strength, functional ability, and inflammatory cytokines when compared with the group with the resistance training and without the supplementation [146].

The supplementation of n-3 FA did not protect from the exacerbation of the disease among the patients suffering from CD [147]. Additionally, n-3 FA did not change a concentration of IL-6, IL-2, IL-5, IL-10, monocyte chemoattractant protein-1, and interferon-gamma, although it decreased IL-1β and IL-4 in patients with CD. Moreover, the supplementation decreased the disease activity, as it reduced CRP (C-reactive protein) and CDAI (Crohn’s Disease Activity Index), and increased the level of hemoglobin [148]. On the other hand, 72% of patients with active UC reduced the medicine dose after n-3 FA supplementation. Additionally, the production of mucosal leukotriene B4 was reduced [149]. According to the study, the consumption of fish and fish products, which are the primary sources of n-3 FA, was similar among UC patients and the healthy men [125].

## 8. Sarcopenia and Physical Activity

The study demonstrated that resistance training performed 2–3 times weekly for 45–60 min increased muscle mass, muscle strength, and muscle fibers, and improve synthesis of muscle proteins [2]. Additionally, the skeletal muscle mass significantly increased in physically active subjects when compared with subjects who used a protein supplement. On the other hand, increased muscle mass was significantly higher in the group which exercised physically and used supplements than in the subjects who only performed exercises [3]. According to Maltais et al., resistance training increased muscle mass and strength regardless of protein intake [150]. Moreover, physical activity, in particular of highintensity and high frequency, elevated muscle strength and BMD [151]. Crucially, free fat mass was not associated with the habitual activity, as there were no significant differences between groups with various levels of habitual activity [152]. Furthermore, the study showed that the time of highintensity of physical activity was connected with an increased grip strength among men, but not among women [153]. The resistance training reduced CRP level and a tendency to a decrease in IL-6 was observed. Thus, it is possible thatdecreased inflammation is due to increased muscle mass [154]. It has been suggested that even short-time resistance training improves muscle strength and speed walking. Hence, according to the recommendation, patients suffering from sarcopenia should perform 20–30 min of aerobic and resistance training sessions three times a week [135].

Patients with IBD often refrain from physical activity, due to fatigue, gastrointestinal symptoms, and limitations stemming from surgical operations. Gatt et al., revealed that physical activity significantly decreased following the diagnosis of IBD. Additionally, UC patients more often resign from physical activity than patients with CD [155]. However, patients with a moderate activity of IBD may regularly do endurance training without risk of disease symptoms [156]. Moreover, in terms of children with IBD the diagnosis did not interfere with playing sports, as only 18% of children could not participate in the preferred sport discipline [157]. It is vital to notice that physical activity may improve social well-being and constitute an additional therapeutic factor [156]. In fact, a three-week low-intensity physical activity intervention based on walking improved the quality of life among patients suffering from CD [158].

## 9. Nutritional Support and Pharmacology in Sarcopenia and Osteoporosis—Future Perspectives

Pharmacotherapy of sarcopenia should be multifaceted. Furthermore, it may be in addition to the therapy of osteoporosis. However, the treatment of an underlying disease which causes sarcopenia is of essence [4]. In the case of low vitamin D serum concentration, the supplementation of these vitamins is crucial. Additionally, patients with nutritional deficiencies require an adequate calorie and protein intake (leucine, beta-hydroxy-beta-methylbutyrate, and their derivatives seem promising) [159,160,161].

There are no specific therapies dedicated to osteosarcopenia, however, recent reports on denosumab seem promising. Denosumab is a human monoclonal antibody (IgG2) directed against the Receptor Activator for Nuclear Factor κ B Ligand (RANKL). A study by Bonnet et al., compared the three-year use of zolendronic acid and denosumab in women with sarcopenia, which showed that patients receiving RANKL inhibitor had a significant increase in lean body weight as well as a significant increase in grip strength [162].

The observations from a nonrandomized study on 79 elderly people reporting for the risk assessment of falls and fractures also seem promising, where the use of denosumab improved the balance and physical performance of patients and contributed to the reduction of the fear of falls [163].Additionally, anabolic steroids and testosterone are useful in pharmacotherapy of sarcopenia [131,132]. These medications have little positive effect on muscle strength and mass, although their use is limited due to adverse effects, including increased risk of prostate cancer in men, virilization in women, and increased cardiovascular risk in both women and men [164,165]. Growth hormone (GH) and insulin-like growth factor IGF-1 did not demonstrate lasting effects on improving muscle mass and muscle function, and their use is further hampered by numerous side effects, such as joint pain, edema, gynecomastia and hyperglycemia [166]. Hence, in the future, probably antibodies of myostatin [133], antibodies of activin receptors [134], and ghrelin agonists (anamorelin) [135] will be used instead. Selective androgen receptor modulators (SARMs) have also been identified by researchers in the context of sarcopenia treatment, and it is suggested that their use may result in an increase in skeletal muscle mass and strength without the androgen-specific side effects [167,168].Other investigated compounds which have a potential use in the treatment of sarcopenia include melatonin, angiotensin converting enzyme inhibitors, eicosapentaenoic acid, thalidomide, celecoxib, VT-122, MT-102, bimagrumab, and ruxolotinib [54,169]. In the experimental methods, CRISPR therapy [170] and stem cell therapy [171] may be important in the treatment of sarcopenia. The potential effect of herbal supplements on the muscle mass of patients with sarcopenia is also emphasized; the researchers further investigate curcumin (*Curcuma longa*), catechins, proanthocyanidin (grape seeds) and gingerols (*Zingiber officinale*) [54].

The aim of the treatment of osteoporosis is to reduce the risk of bone fracture and improve the quality of life [172].The treatment consists of supplementing calcium and vitamin D deficiencies, which are effective medications in osteoporosis, prevention of falls and contributing to a healthy lifestyle. The medications used in the treatment of osteoporosis comprise antiresorption medications (bisphosphonates, denosumab, hormone replacement therapy, selective estrogen receptor modulators), anabolic medications (teryparatide), mixed mechanism medications (strontium ranelate). The prevention of osteoporosis focuses on an appropriate diet, regular exercise, maintaining proper body weight, avoiding alcohol and cessation of smoking [18,173].

## 10. Summary and Conclusions

Osteosarcopenia is a disorder associated with a loss of muscle and bone tissues, diagnosed when sarcopenia and osteoporosis occur simultaneously. Sarcopenia comprises a gradual decrease in muscle mass with a progressive loss of muscle strength. Osteopenia and osteoporosis are diagnosed on the basis of a reduced BMD. Both sarcopenia and osteoporosis, which is preceded by osteopenia, are strongly connected due to a close correlation between the skeletal and muscular systems. Sarcopenia and osteoporosis increase the risk of falls, fractures, resulting in a decreased quality of life, and may cause disability or even death. Both syndromes affect patients with chronic diseases, such as patients suffering from IBD at any age. The treatment of osteosarcopenia is an important element of therapy of IBD patients, and it requires either physical activity or adequate nutrition. Nevertheless, there are numerous substances under clinical trial pending as a standard therapy of both osteoporosis and sarcopenia [167,168].

## Figures and Tables

**Table 1 nutrients-13-00293-t001:** Algorithm of the sarcopenia diagnosis.

Stage of Sarcopenia Algorithm	Using Methods
F—find cases	SARC-F (strength, assistance with walking, rise from a chair, climb stairs, and falls) questionnaire or screening test Ishii [13,14,15]
A—assess	assess muscle strength (grip strength using a dynamometer) [16] or perform the chair stand test [2]
C—confirm	based on the low quality or quantity of muscle using DXA (dual-energy X-ray absorptiometry), BIA (bioelectrical impedance analysis), CT (computed tomography), MRI (magnetic resonance imaging) [2]
S—severity	evaluated by various tests, such as SPPB (short physical performance battery), TUG test (timed-up and go test), 400 m walk test, the stair climbing test [2]

**Table 2 nutrients-13-00293-t002:** Risk factor characteristics of sarcopenia in patients suffering from IBD.

ARisk Factor of Sarcopenia	Characteristics
Low level of physical activity	Sedentary lifestyle, decreased physical activity, zero gravity [2,44].
Hormones and cytokines imbalance	Age-related decreased level of testosterone, growth hormone, thyroid hormones, insulin-like growth factor (IGF).
	Increased proinflammatory cytokines (TNFα-tumour necrosis factor α, IL-6) which inhibit the anabolic and elevate the catabolic processes [45].
Protein synthesis	Decreased ability to protein synthesis combined with an insufficient energy intake; accumulation of lipofuscin and noncontractile material in the muscle [45].
Motor unit remodeling	Decreased number of motor units, a disorder of satellite cells activation and their differentiation to form skeletal muscle fibers [55].
Evolution factors	Contemporary lifestyle, combining life prolongation and sedentary lifestyle [56].
Impact of early development stages	Impact of environmental factors on the parenteral development—low birth weight as a factor of nutrient deficiency delivered to fetus, which leads to decreased muscle strength and muscle mass in adult life [57,58,59].

**Table 3 nutrients-13-00293-t003:** The risk factors of osteoporosis inIBD patients [65].

Risk Factors of Osteoporosis	Characteristics
Malnutrition	A nutritional state in which deficiency (or excess) of energy, protein and micronutrients affects tissues or/and body form and function and clinical outcome [66,67,68].
Low BMI	<18.5 kg/m^2^ [66,67].
Malabsorption	Changes in gastrointestinal tracts, affecting digestion, absorption and transport of nutrients; important factors may include lactose intolerance, which is common among IBD patients [67,68].
Decreased physical activity	Physical activity is increased energy expenditure due to the skeletal muscle contractions associated with body movement [69].
Chronic inflammation	Inflammation lasting over 6 weeks; IBD I associated with a higher concentration of proinflammatory cytokines, such as Il-9, Il-17, TNF-alpha [70,71,72,73].
Nutritional deficiency	e.g., vitamin D, calcium, magnesium [65,74,75].
Hormonal disorders	e.g., hypogonadism, thyroid disorders [76,77].
Genetic factors	Certain genes constitute candidate genes associated with osteoporosis: vitamin D receptor (VDR) gene, estrogen receptor (ESR1) gene, bone morphogenetic protein (BMP2), osteoprotegerin (OPG) gene, transforming growth factor beta (TGF-β) gene [78,79,80,81,82].
Age	Over 50 years of age [83].

**Table 4 nutrients-13-00293-t004:** Calorie, protein, fat, and vitamin intake and the risk of osteosarcopenia in patients suffering from IBD.

Element of the Diet	Comments
Energy	Sufficient or higher demand.
	Malnutrition should be avoided since it is a risk factor of osteoporosis and sarcopenia [65,104].
Protein	1.2–1.5 g/kg body weight for patients suffering from sarcopenia [113].
Fat	Omega-3 fatty acids may affect muscle tissue, although data concerning the impact of n-3 fatty acids on bone and muscle tissue are unclear [65,114].
Vitamin D	Vitamin D affects the proliferation and differentiation of muscle cells. Moreover, a deficiency of vitamin D is a risk factor of osteoporosis [65,115].

## Data Availability

Statement excluded.

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
