# Peer review of "A Vicious Cycle of Osteosarcopenia in Inflammatory Bowel Diseases—Aetiology, Clinical Implications and Therapeutic Perspectives"

_nutrients, 2021, doi:10.3390/nu13020293_

Round 1

Reviewer 1 Report

I have really appreciated the efforts put into tackling such delicate subject.

In spite of this, the manuscript presents several issues that should be fixed prior considering its publication.

  • First, English language is understandable, but the main text is discontinuous, very repetitive, and full of lexical mistakes. Unfortunately, the language barrier is not negligible, and a clear and concise exposition might supply a larger amount of information than a formal and redundant report.
  • The second main issue is the disconnection between the title of the manuscript and the actual content. The title actually promises a dynamic discussion about the current state of the art, and an exciting evaluation of possible future perspectives. The text, instead, lies flat over the description of the literature. The paragraphs are mostly detached and independent from each other. Furthermore, the discussion is not really a discussion, excessively brief and unsubstantial.
  • This is just a suggestion, but maybe a few more tables might help cut down the redundancy and concisely deliver an effective message.

A few formatting issues are present. I am not sure about the meaning of a sudden question mark in the introduction paragraph “(?)”, and the paragraphs about Vitamin D and n-3 fatty acids are erroneously positioned beneath the “protein” paragraph, as if they belonged to the protein category. In addition to this, the style adopted for the titles of the paragraphs is inconsistent, it shifts from a description of the content to a mere placeholder.

Reviewer 2 Report

I read the manuscript entitled "How to break a vicious cycle of osteosarcopenia in inflammatory bowel disease? Dietary prevention and treatment" with interest, considering pathophysiological characteristics of IBD contributing muscle loss. Even though I recognized that the manuscript includes a lot of relevant issues regarding sarcopenia and IBD, there are still issues to be improved for publication

Major points

  1. Components and paragraphs of the manuscript are rather detached, without some tangible logical linkages between paragraphs and units. Especially, General knowledge on sarcopenia and IBD specific issues should be more well blended.
  2. While the title includes 'Osteosarcopenia', prevention/treatment part lacks any in-depth description on 'osteo' issue in IBD
  3. There are tons of awkward English writings, requiring extensive English editing with logical consideration

Minor points

  1. I recommend authors rather not to use the term elderly in the scientific literature, (See Vaughan et al, JAGS, 2019 67:211-217), and replace it with case-specific terms such as older people, older participants, individuals etc. 
  2. Typos including "anty-TNFa" at line 179, "unproperly diet" at line 181, "there affect" at line 267.
  3. "Following the recommendation, the intake of 50,000IU vitamin D weekly is safe, line 255" -- can we confirm that this is absolutely safe, even with limited available evidence? There are many claims in the manuscript with this kind of overstate/generalization. 

Round 2

Reviewer 2 Report

I found the revised manuscript substantially improved from the original one, especially in terms of readability.